# Characterization of a human monoclonal antibody generated from a B-cell specific for a prefusion-stabilized spike protein of Middle East respiratory syndrome coronavirus

Jang-Hoon Choi[1], Hye-Min Woo[2], Tae-young Lee[2], So-young Lee[2], Sang-Mu Shim[2], Woo-Jung Park[2], Jeong-Sun Yang[2], Joo Ae Kim[3], Mi-Ran Yun[3], Dae-Won Kim[3], Sung Soon Kim[4], Yi Zhang[5], Wei Shi[5], Lingshu Wang[5], Barney S. Graham[5], John R. Mascola[5], Nanshuang Wang[6], Jason S. McLellan[6], Joo-Yeon Lee[2], Hansaem Lee[2]*

1 Division of Viral Disease Research, Center for Infectious Diseases Research, Korea National Institute of Health, Korea Centers for Disease Control and Prevention, Cheongju-si, Republic of Korea, 2 Division of Emerging Infectious Disease and Vector Research, Center for Infectious Diseases Research, Korea National Institute of Health, Korea Centers for Disease Control and Prevention, Cheongju-si, Republic of Korea, 3 Division of Vaccine Research, Center for Infectious Diseases Research, Korea National Institute of Health, Korea Centers for Disease Control and Prevention, Cheongju-si, Republic of Korea, 4 Division of Bacterial Disease Research, Center for Infectious Diseases Research, Korea National Institute of Health, Korea Centers for Disease Control and Prevention, Cheongju-si, Republic of Korea, 5 Vaccine Research Center, National Institute of Allergy and Infectious Diseases, National Institutes of Health, Bethesda, MD, United States of America, 6 Department of Molecular Biosciences, College of Natural Sciences, University of Texas, Austin, TX, United States of America

* saem777@korea.kr

## Abstract

Middle East respiratory syndrome coronavirus (MERS-CoV) causes severe respiratory infection and continues to infect humans, thereby contributing to a high mortality rate (34.3% in 2019). In the absence of an available licensed vaccine and antiviral agent, therapeutic human antibodies have been suggested as candidates for treatment. In this study, human monoclonal antibodies were isolated by sorting B cells from patient's PBMC cells with prefusion stabilized spike (S) probes and a direct immunoglobulin cloning strategy. We identified six receptor-binding domain (RBD)-specific and five S1 (non-RBD)-specific antibodies, among which, only the RBD-specific antibodies showed high neutralizing potency (IC$_{50}$ 0.006–1.787 μg/ml) as well as high affinity to RBD. Notably, passive immunization using a highly potent antibody (KNIH90-F1) at a relatively low dose (2 mg/kg) completely protected transgenic mice expressing human DPP4 against MERS-CoV lethal challenge. These results suggested that human monoclonal antibodies isolated by using the rationally designed prefusion MERS-CoV S probe could be considered potential candidates for the development of therapeutic and/or prophylactic antiviral agents for MERS-CoV human infection.

**Data Availability Statement:** All relevant data are within the manuscript and its Supporting Information files.

**Funding:** This study was supported by a Korea National Institute of Health fund (2016-NG47001-00, 4845-300-210-13) and intramural funding from the National Institute of Allergy and Infectious Diseases, NIH, USA. J. C., S. S. K., B. S. G., J. R. M., L.W. and H. L. designed the experiments. J.C., L.W., H.M.W, and H.L. expressed and purified antibodies. J. C. and H. L. prepared this manuscript. H. M. W., J. A. K. and W. J. P. carried out the neutralizing assay with live MERS-CoV. Y. Z. and W. S. expressed and purified RBD, S1, and S2 proteins. N. W. and J. M. designed and produced the S-2P probe. T. L., S. L., S. M. S. and J. S. Y. performed the animal experiments. M. R. Y. and D. W. K. analyzed the computational structure modeling of RBD and mAb interation. J.C. and H.L. wrote the paper with contributions from S.S.K, L. W. and B. S. G.

**Competing interests:** We also state that competing interests: the patent name "MONOCLONAL ANTIBODY FOR SPIKE PROTEIN OF MIDDLE EAST RESPIRATORY SYNDROME CORONAVIRUS, AND USE THEREOF" and No. 10-1895229 (Korean patent) (PCT application No. PCT KR2018009754). This does not alter our adherence to PLOS ONE policies on sharing data and materials.

# Introduction

Since the first report of a case of Middle East respiratory syndrome coronavirus (MERS-CoV) in Saudi Arabia in 2012, 2,519 laboratory-confirmed patients and 866 deaths (34.3% mortality) have been reported as of January 2020 [1]. Laboratory confirmed cases have been reported in 27 countries globally, most of which were caused by human-to-human transmission in household or healthcare settings besides cases of primary infection [2]. The largest outbreak outside of the Arabian Peninsula occurred in South Korea in 2015, with a total 186 confirmed cases and 38 deaths (20.4% mortality) [3].

No vaccine or therapeutic agent has been approved for the prevention and treatment against MERS-CoV infection to date. Several laboratory studies proposed vaccine candidates, based on the receptor-binding domain (RBD) of spike (S) antigen, which induce neutralizing antibodies. These antibodies inhibit RBD from binding the host receptor dipeptidyl peptidase 4 (hDPP4; also known as CD26) [4], An alternative strategy is to use virus-specific therapeutic human antibodies that can be rapidly developed to neutralize the virus during infection. Indeed, some monoclonal antibodies (mAbs) cloned from humans using various platforms have shown considerable MERS-CoV-neutralizing potency along with therapeutic or prophylactic efficacies in animal models [5–12]. Additionally, nanobodies and camel/human hybrid antibodies have been suggested for treatment of viral infection [13, 14]. Most recently, a research group isolated mAbs from a MERS patient via B cell immortalization, which showed promising therapeutic and prophylactic potential against MERS-CoV infection in a mouse model [8]. Moreover, targeting multiple antigenic sites on spike might be advantageous, suggesting the possibility of combination therapy with RBD-specific antibodies for escape-mutant prevention [11, 15]. To date, two phase I human clinical trials of therapeutic antibody candidates, humanized mAbs (REGN3048, REGN3051) and human polyclonal anti-MERS immunoglobulin G (IgG; SAB-301) from transchromosomic cattle have been completed [16, 17]. The clinical trial result of SAB-301 shows that SAB-301 appears to be safe and well-tolerated in healthy individuals, encouraging expectations for the development of MERS therapeutics [16].

Despite two decades of extensive research and development of therapeutic antibodies against viral infections (numerous clinical trials have been performed for both acute and chronic infections), only palivizumab and ibalizumab have been approved for monoclonal prophylaxis of human respiratory syncytial virus and human immunodeficiency virus infections, respectively [18, 19]. Several strategies are available for the development of therapeutic human mAbs for viral infections; in the present study, we isolated single B cells from patients for development. A MERS-CoV S antigen trimer probe (S-2P) was designed to retain betacoronavirus S proteins in the prefusion conformation [20]. It increased antigenicity and affinity binding to hDPP4. It also elicited high titers of neutralizing antibodies in mice. Previous reports, including MERS-CoV, SARS-CoV, HIV and RSV, showed that structure-designed stabilization of the viral surface protein by mutations induced greater neutralizing mAbs than postfusion proteins [20–23].Thus, we rationally designed a strategy to develop the potent neutralizing antibodies against MERS-CoV infection by using an S-2P. We directly isolated B cells from the peripheral blood mononuclear cells (PBMCs) of convalescent patients using an S-2P for cell sorting, cloned the IgG genes, and produced antibodies through cell culture [15]. We also evaluated the binding affinity for S protein and the RBD of the obtained mAbs and determined the neutralizing ability with a pseudovirus and plaque- reduction neutralizing assay (PRNT) *in vitro*. Finally, we examined the therapeutic efficacy of the mAb with the strongest neutralizing activity in transgenic (TG) mice challenged with a lethal dose of MERS-CoV. These results offer a new platform and antibody for prompt development toward clinical trials.

## Materials and methods

### Clinical specimens and ethics statement

PBMCs and serum were collected from three convalesced MERS-CoV Korean patients 1 month after diagnosis and deposited at the National Human Resource Bank (NHRB). Cryo-preserved specimens were obtained from the NHRB after the research protocols were approved by the Korea Centers for Disease Control and Prevention Institutional Review Board (IRB No. 2016-05-01). All specimens were fully anonymized.

### Single B cell isolation by flow cytometry

MERS-CoV S-reactive single B cells were sorted into 96-well plates as previously described [15]. Briefly, cryopreserved PBMCs were thawed in a 37˚C water bath, added to 20 ml of wash buffer [phosphate-buffered saline (PBS) containing 10% fetal bovine serum and 2 U/ml DNase I] and centrifuged at 200 g for 8 min. The cells were resuspended with 10 ml of PBS and placed on a 40-μm nylon cell strainer (BD Bioscience, San Jose, CA, USA) to remove aggregated cells. After cen-trifugation, cells were stained with a LIVE/DEAD Fixable Violet Dead Cell Stain Kit (Life Tech-nologies, Carlsbad, CA, USA) and incubated with 200 μl of an anti-human antibody cocktail, including CD3 (BV421; BD Bioscience), CD4 (BV421; BioLegend, San Diego, CA, USA), CD8 (BV421; BioLegend), CD14 (BV421; BioLegend), CD20 (PerCP Cy5; BioLegend), and IgG (FITC; BD Bioscience) antibodies, along with PE-labeled S-2P trimer [20] and APC-labeled S1 for 20 min in the dark. Except for B cell, other immune cells were excluded by negative selection, then S-2P trimer- and/or S1 probe-positive single B cells were sorted by flow cytometry on a FACSAria III cell sorter (BD Bioscience) and then directly collected into a 96-well polymerase chain reaction (PCR) plate containing RNase OUT (Invitrogen, Carlsbad, CA, USA), 5× First-Strand Buffer (Invitrogen), 0.1 M dithiothreitol (Invitrogen), and Igepal (Sigma-Aldrich, St Louis, MO, USA).

### Reverse transcription (RT) and nested PCR for human IgG genes

RT was performed directly by adding a random hexamer primer (Invitrogen), 10 mM dNTP (Invitrogen), and reverse transcriptase Superscript III (Invitrogen) in 96-well PCR plate and incu-bated as follows: 42˚C for 10 min, 25˚C for 10 min, 50˚C for 50 min, and 94˚C for 5 min. The var-iable regions of Ig heavy and light-chain genes were amplified using primers as reported previously [24]. For the first-round of PCR, 5 μl of cDNA templates were added to the reaction mixture containing 5 μl 10× PCR buffer, 1 μl 25 mM $MgCl_2$, 0.5 μl 10 mM dNTP (Fermentas, Waltham, MA, USA), 1 μl of forward and reverse primers (25 μM each), 0.3 μl HS Taq Plus Poly-merase (Qiagen, Hilden, Germany), and 36.2 μl nuclease-free water. The PCR cycle was initiated by 5-min incubation at 94˚C, followed by 15 cycles of 94˚C for 30 s, 51˚C for 30 s, 72˚C for 1 min, and 30 cycles of 94˚C for 30 s, 56˚C for 30 s, and 72˚C for 1 min. Subsequent nested PCR was per-formed with a reaction mixture containing 3 μl of the first PCR amplicon, 5 μl 10× PCR buffer, 10 μl solution Q (Qiagen, Hilden, Germany), 0.5 μl 10 mM dNTP, 1.5 μl forward and reverse primers (25 μM each), 0.3 μl HS Taq Plus Polymerase, and 28.2 μl nuclease-free water. The second PCR cycle was initiated by 4-min incubation at 94˚C, followed by 50 cycles of 94˚C for 30 s, 57˚C for 30 s, 72˚C 30 s, and final extension at 57˚C for 30 s. PCR products were visualized on 2% aga-rose gels, and positive samples were analyzed by sequencing with 3′-end primers.

### Cloning and expression of IgG

Paired heavy and light-chain genes were cloned into an expression vector, and IgGs were expressed from transfected Expi293 cells as described previously [15]. Briefly, each heavy and light-chain sequence was evaluated using the IMGT-QUEST tool which provides single chain

fragment variable and each sequence productivity [25]. Then, the variable regions of heavy (VH) and light (VK or VL) chains were cloned into expression vectors (VRC2271, VRC2272 and VRC2276) containing constant regions of human immunoglobulin heavy (γ), light κ, and λ chains, respectively. Same amounts of heavy and light-chain vectors were co-transfected into Expi293 cells using the ExpiFectamine 293 transfection kit (Thermo Fisher Scientific, Carlsbad, CA, USA) according to manufacturer instructions. Cells were incubated for 4 to 5 days at 37˚C and 8% $CO_2$; subsequently, cells were centrifuged at 2500 g, and supernatants were harvested. IgG was purified using protein A Sepharose 4 Fast Flow (GE Healthcare, Uppsala, Sweden).

### Enzyme-linked immunosorbent assay (ELISA)

The 96-well Maxisorp plates (Nunc, Roskilde, Denmark) were coated with a previously reported recombinant MERS-CoV S-2P trimer [20], RBD, S1, or S2-mFc (S2-fusion with murine Fc for solubility and stability) protein [15] (0.1 μg/well) in PBS at 4˚C overnight. Subsequently, plates were blocked with blocking buffer (5% skim milk and 2% bovine serum albumin in PBS) at room temperature for 1 h and washed three times with PBS containing 0.05% Tween-20. Then, serial 4-fold-diluted mAbs were added to each well, incubated for 1 h, and washed three times with PBS containing 0.05% Tween-20. Horseradish peroxidase-conjugated anti-human IgG (Jackson Laboratory, Bar Harbor, ME, USA) was used as a secondary antibody. The color signal was developed by tetramethylbenzidine (KPL, Gaithersburg, MD, USA), and the reaction was stopped by adding 1 N $H_2SO_4$ (Thermo Fisher Scientific). Absorbance at an optical density of 450 nm was measured using a microplate reader (Molecular Devices, Sunnyvale, CA, USA).

### Pseudovirus-neutralization assay and PRNT

Pseudovirus-neutralization assay (for EMC) with luciferase activity [15] and PRNT assay (for EMC and MERS-CoV/KOR/KNIH/002_05_2015 (KNIH-002) strains) were performed as described previously [26]. Briefly, serial 4-fold-diluted mAbs and an equal volume of virus (40 pfu/well) were incubated at 37˚C for 2 h. The mixture was added to each well of a 24-well plate seeded with Vero cells ($1 \times 10^5$ cells/well) and incubated at 37˚C for 1 h, followed by addition of 1 ml of 1.5% carboxymethylcellulose on the medium. After 3 to 4 days of incubation, the cells were fixed with 4% paraformaldehyde and stained with crystal violet to visualize the plaques. All assays were performed in triplicate for each mAbs. The half-maximal inhibitory concentration ($IC_{50}$) of each mAb was calculated from dose-response plots using GraphPad Prism software (GraphPad, La Jolla, CA, USA).

### Binding affinity of the mAbs to S1 and RBD

The binding affinities of the mAbs to MERS-CoV S antigen were determined by a surface plasmon resonance (SPR) assay with a Biacore T200 instrument (GE Healthcare) [27]. Briefly, purified MERS-CoV S1 or RBD protein was immobilized onto a CM5 chip in 1× PBS buffer (pH 7.4). Different concentrations of RBD-specific mAbs (KNIH77-A5, KNIH77-A6, KNIH90-B2, KNIH90-B7, KNIH90-F1, and KNIH90-F2) were run in HBS-EP buffer. The surface was regenerated with 10 mM NaOH or 10 mM glycine-HCl (pH 2.0). Sensograms were fit with a 1:1 binding model using BIA Evaluation software (GE Healthcare).

### Computational structure modeling

The RBD (positions 367–606 in the full S protein) of the S protein was modeled using the I-TASSER web service (zhanglab.ccmb.med.umich.edu/I-TASSER) [28]. The variable region

structure of the human neutralizing antibody KNIH90-F1 was generated by Rosetta Antibody modeling in ROSIE (http://rosie.graylab.jhu.edu/antibody) [29]. Docking simulation with the RBD/KNIH90-F1 complex was performed using the SnugDock program in ROSIE [30]. Modeled structures of the RBD and KNIH90-F1 were used as the starting structures for the docking study. Structural alignment and visualization were performed with the PyMOL program (The PyMOL Molecular Graphics System, v.2.0; Schrodinger, LLC, New York, NY, USA).

## Mice and ethics statement

All animal experiments were performed in strict accordance with the Guidelines for the Care and Use of Laboratory Animals of Korea National Institute of Health (KNIH). The protocol was approved by the Institutional Animal Care and Use Committee of the KNIH (Protocol approval No. KCDC-090-18-2A). C57BL/6N mice were purchased from Orient Bio (Iksan, Republic of Korea). Cages were changed once per week until the end of the experiment, and food and water replenished as often as was necessary.

## Assessment of therapeutic and prophylactic effects of mAb in mice

The wild-type MERS-CoV animal challenge experiment was performed under animal biosafety level 3 containment and practice. Ten-week-old male C57BL/6N mice expressing hDPP4 (C57BL/6N-hDPP4) conjugated with a FLAG tag were generated according to the method previously described for developing a TG mouse expressing hDPP4 [31]. Prior to lethal challenge, 50% mean lethal dose ($LD_{50}$) in mice was determined according to the previously described method [32]. $LD_{50}$ of MERS-CoV KNIH-002 is $1 \times 10^3$ pfu per mouse. Five mice were inoculated with a 100-fold $LD_{50}$ ($1 \times 10^5$ pfu) of MERS-CoV KNIH-002 strain via intranasal route, whereas three different doses (8 mg/kg, 2 mg/kg, and 0.4 mg/kg; n = 9 per dose) of the mAb (KNIH90-F1) or PBS were administered intraperitoneally on day 1 post-infection. Four mice in the group treated at the highest dose (8 mg/kg) were humanely euthanized by intraperitoneal injection using a mix of 30 mg/kg Zoletil® (Virbac, Carros, France) and 10 mg/kg Rompun® (Bayer, Leverkusen, Germany) on day 4 post-infection, and lungs and brains were harvested for analysis of MERS-CoV replication. After brain and lung tissue weights were measured, tissue samples in 1 mL of DMEM media were homogenized, and MERS-CoV was titrated by plaque assay and real-time RT-PCR. To measure the virus titer, tissues were homogenized in DMEM using a manual homogenizer and centrifuged for 10 min. Viral RNAs were extracted from the supernatant by Maxwell RSC Viral TNA kit (Promega) and were assayed by using PowerChek™ MERS (upE & ORF1a) Real-time RT-PCR kit (Kogene biotech) according to the manufacturer's instructions. To generate the standard RNA of upE, upE RT-PCR was performed with SuperScript III reverse transcriptase (Invitrogen, Carlsbad, CA, USA) using MERS-CoV EMC as an RNA template. For PCR, 5 μl of the RT reaction was added to 20 μl of PCR amplification master mix containing Ex-Taq™ polymerase (Takara, Shuzo, Japan). After PCR purification with a PCR purification kit (Qiagen), 1 μl of pCR®2.1-TOPO® vector (Invitrogen) was added to the purified PCR products, and the mixture was transformed into One Shot® Top10 competent cells (Invitrogen). The standard RNAs were synthesized massively using a MEGAscript® T7 kit (Applied Biosystems) and the template plasmids pCR2.1-upE. The RNAs were purified by MEGAclear™ kit (Applied Biosystems). RNA copy numbers were calculated based on RNA concentration and size. The upE RNA fragment was diluted serially with a 0.5 log-dilution to yield 1 to 105 copies per reaction, and these dilutions were then used as templates for rRT-PCR. Then, the Ct values of tissues and copy numbers were calculated.

Vero cell monolayers were inoculated with serial 10-fold dilutions of the supernatant, which was adsorbed for 1 hr. The cells were covered with 1 ml or 1.5% CMC solution. After 3 days, the cells were washed with PBS and fixed with 4% formaldehyde and stained with crystal violet. The body weight of the remaining five mice in each group was monitored daily for 14 days, and mice showing a weight loss of more than 25% of their initial body weight were euthanized by $CO_2$ gas and recorded as dead. The remaining five mice in each group were observed for 14 days for survival analysis. To evaluate the prophylactic effect, an additional four mice were pretreated with 8 mg/kg of the mAb 1 day before the challenge, and viral replication in lung and brain tissues was determined on day 4 post-infection.

## Histology

Six mice in each group were performed according to the same experiment design for the therapeutic efficacy in the previous section. Three mice in each group were humanely euthanized on days 4 and 7 post-infection, and lung and brain tissues were harvested for histopathological evaluation. The tissues were fixed in formalin, embedded in paraffin blocks, sectioned, followed by hematoxylin and eosin (H&E) staining.

## Real-time RT-PCR for virus detection

MERS-CoV viral RNAs were detected by one-step real-time RT-PCR as previously described [33]. Briefly, viral RNAs were extracted from homogenized mouse tissues using the Maxwell RBC RNA TNC kit (Promega, Fitchburg, WI, USA). Real-time RT-PCR was conducted with the PowerChek MERS assay (Kogenebiotech, Seoul, Republic of Korea) to detect *Orf1a* genes according to manufacturer instructions. Five microliters of template RNA was added to the reaction mixture, and the following PCR cycling program was used: 50˚C for 30 min, followed by 95˚C for 10 min and 40 cycles of at 95˚C for 15 s and 60˚C for 60 s.

## Statistical analysis

Group comparisons were performed by a paired Student's *t* test in Prism 7 (GraphPad Software, San Diego, CA, USA). Differences were considered significant at $P < 0.05$.

## Results

### Generation and binding specificity of MERS-CoV mAbs

We separated MERS-CoV S-binding single B cells from the PBMCs of three convalescent Korean MERS patients using flow cytometry cell sorting. IgG heavy and light (kappa or lambda) chain variable regions were amplified by nested PCR from 90 sorted B cells (24 cells from a patient and 66 cells from pooled two patient's PBMCs), cloned into expression vectors, and expressed in Expi293 cells after transient transfection, followed by purification of 11 IgGs from culture supernatants using protein A beads. The total protein yields ranged from 1 mg/mAb to 25 mg/mAb per 200-ml culture. Evaluation of the binding specificities of the purified IgGs by ELISA coated with soluble S-2P trimer, S1, RBD, and S2-mFc antigens revealed that six antibodies (KNIH77-A5, KNIH77-A6, KNIH90-E5, KNIH90-E6, KNIH90-F1, and KNIH90-F2) bound to S-2P trimer, S1, and RBD and were thus considered to be RBD-specific (Fig 1). The other five antibodies (KNIH90-A3, KNIH90-A9, KNIH90-B2, KNIH90-B7, and KNIH90-C4) bound to the S-2P trimer and S1 but not RBD; however, we identified no S2 domain-specific antibody, as this domain has relatively low immunogenicity as compared with the variable S1 domain. Only KNIH90-C4 bound to prefusion S trimer more strongly than S1 protein in ELISA (Fig 1B). Also, the specificity of mAbs were tested in immunofluorescence

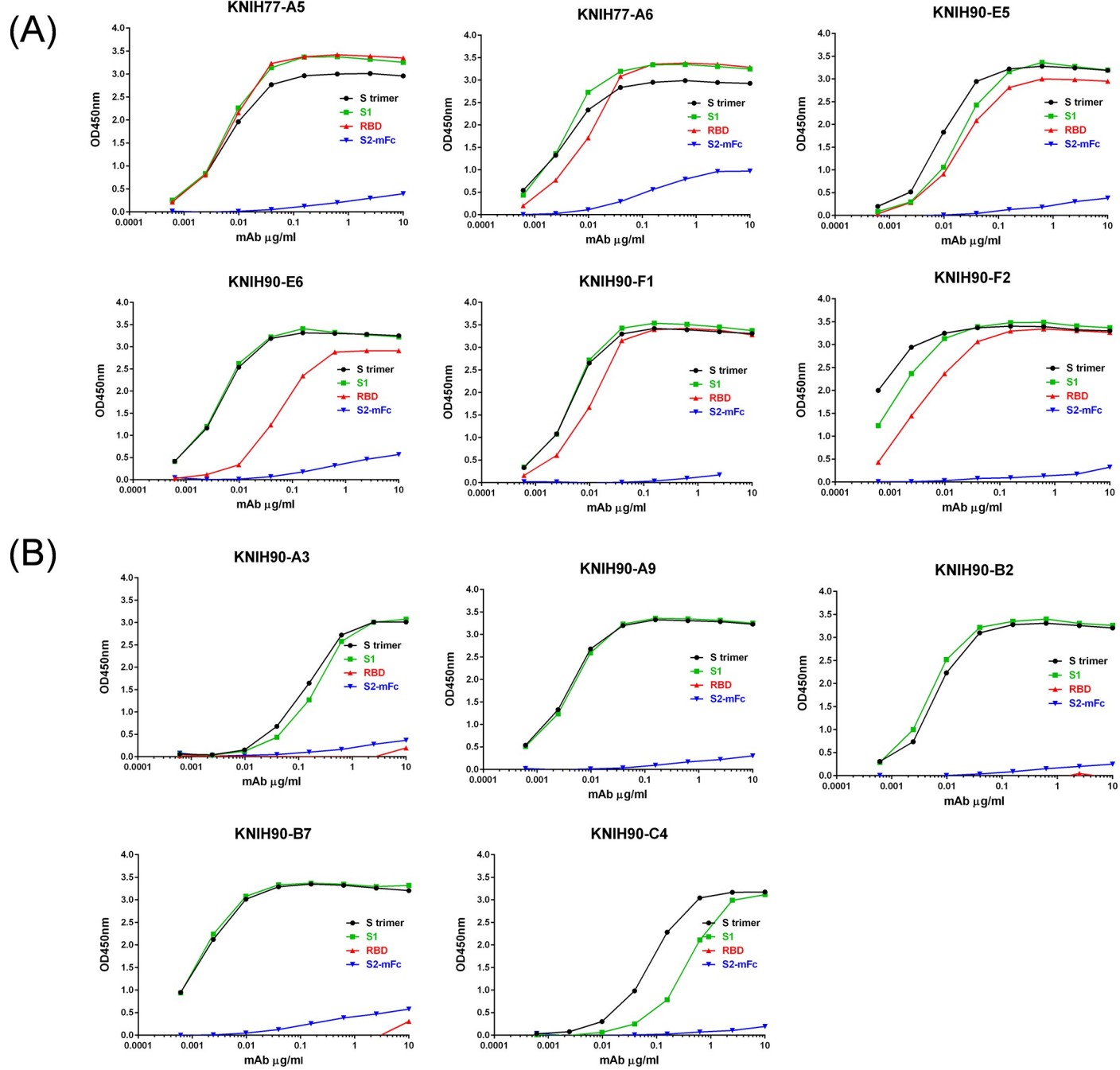

**Fig 1. Binding of human mAbs to S1 and RBD of the MERS-CoV S protein.** Binding specificity of the mAbs was assessed by ELISA with a soluble S-2P trimer, S1, RBD sub-domains, and S2-mFc. Among the 11 purified mAbs, six were specific to RBD and five were specific to S1 (non-RBD). RBD-specific mAbs bound to both RBD and S1 protein. No S2-specific mAbs (blue line) were identified. Data points represent the mean of three technical replicates with standard errors.

assay (IFA) with other coronaviruses, human coronavirus (HCoV)-OC43, HCoV-229E, HCoV-NL63, and SARS-CoV-2. All mAbs were specific for MERS-CoV. All mAbs showed no indication of binding to other coronavirus strains tested (S3 Fig).

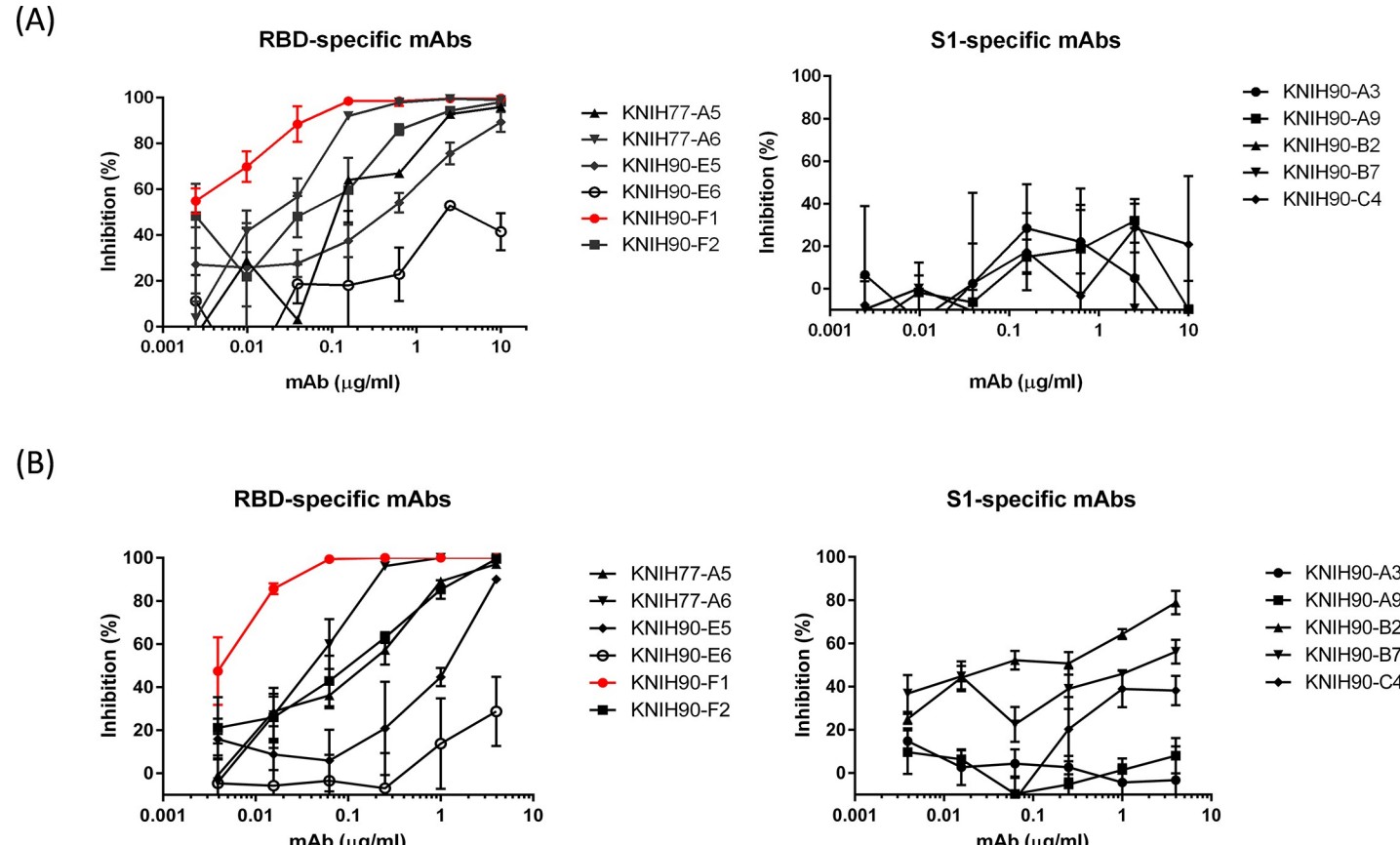

**Fig 2. Neutralizing activity of mAbs specific for S1 and the RBD of the S protein.** Neutralization potency of RBD and S1-specific mAbs evaluated using a MERS-CoV EMC pseudovirus-neutralization assay (A) and a live-virus PRNT assay (B). The percent inhibition by the mAbs was determined by comparison with untreated cells. Each experiment was performed in triplicate, and data represent the mean of each experiment with standard errors.

## Neutralization activity of the mAbs

We initially evaluated the neutralization potency of the mAbs using MERS-CoV (EMC) pseudovirus. Among the 11 purified mAbs, the five S1 (non-RBD)-specific mAbs failed to inhibit pseudovirus particles, but five of the six RBD-specific mAbs showed high neutralization capacity, with KNIH90-E6 failing to reach 50% inhibition, even at the highest mAb concentration tested (Fig 2A). The $IC_{50}$ values for the RBD-specific mAbs varied between 0.2173 µg/ml and 0.0089 µg/ml, with KNIH90-F1 demonstrating the most potent inhibition (Table 1, S1 Fig).

**Table 1. Neutralization activities of RBD-specific mAbs to MERS-CoV EMC strain.**

| mAbs | Live MERS-CoV $IC_{50}$ (µg/ml) | Pseudovirus $IC_{50}$ (µg/ml) |
|---|---|---|
| KNIH77-A5 | 0.075 | 0.1572 |
| KNIH77-A6 | 0.030 | 0.0089 |
| KNIH90-E5 | 1.787 | 0.9456 |
| KNIH90-E6 | N.D. | N.D. |
| KNIH90-F1 | 0.006 | 0.0135 |
| KNIH90-F2 | 0.247 | 0.2173 |

The data are expressed as mean $IC_{50}$ (n = 3). N.D.: Not Determined

**Table 2. Binding affinity of mAbs with potent neutralizing activity to MERS-CoV spike.**

| mAbs | Binding kinetics for S1 | | | Binding kinetics for RBD | | |
|---|---|---|---|---|---|---|
| | Ka (1/Ms) | Kd(1/s) | $K_D(M)$ | Ka (1/Ms) | Kd(1/s) | $K_D(M)$ |
| KNIH77-A5 | $1.33 \times 10^5$ | $\leq 10^{-5}$ | $\leq 7.53 \times 10^{-11*}$ | $1.29 \times 10^5$ | $2.07 \times 10^{-5}$ | $1.61 \times 10^{-11}$ |
| KNIH77-A6 | $6.37 \times 10^4$ | $4.42 \times 10^{-5}$ | $6.94 \times 10^{-10}$ | $4.67 \times 10^4$ | $3.14 \times 10^{-5}$ | $6.73 \times 10^{-10}$ |
| KNIH90-B2 | $1.30 \times 10^5$ | $\leq 10^{-5}$ | $\leq 7.65 \times 10^{-11}$ | - | - | - |
| KNIH90-B7 | $6.81 \times 10^5$ | $1.04 \times 10^{-4}$ | $1.52 \times 10^{-10}$ | - | - | - |
| KNIH90-F1 | $6.48 \times 10^5$ | $9.88 \times 10^{-5}$ | $1.52 \times 10^{-10}$ | $3.76 \times 10^5$ | $2.70 \times 10^{-5}$ | $7.19 \times 10^{-11}$ |
| KNIH90-F2 | $1.37 \times 10^5$ | $8.81 \times 10^{-4}$ | $6.59 \times 10^{-9}$ | $2.38 \times 10^5$ | $1.39 \times 10^{-3}$ | $5.85 \times 10^{-9}$ |

[*] Values that exceeded T200 detection limit $10^{-5} s^{-1}$

Similar results were obtained in the PRNT assay using wild-type MERS-CoV (EMC) virus (Fig 2B). Therefore, we identified KNIH90-F1 as the most potent RBD-specific mAb against MERS-CoV (EMC and KNIH-002) in both the pseudovirus neutralization assay (IC$_{50}$: 0.0135 μg/ml) and PRNT assay (IC$_{50}$: 0.006 μg/ml) (Table 1, S1 Fig). However, in contrast to the pseudovirus neutralization assay, some S1-specific (non-RBD) mAbs (KNIH90-B2 and KNIH90-B7) also moderately inhibited the wild-type virus, although to a lesser extent than the RBD-specific mAbs (Fig 2B). Spike of the KNIH-002 strain has four amino acid changes (S137R, I529T, V530L, and R629H), compared to EMC strain [34]. Among them, I529T and V530L mutations were located in the RBD, but not at the RBD-DPP4 receptor interface [35]. Thus, we assumed that KNIH90-F1 would have similar neutralizing activities and affinities between MERS-CoV KNIH-002 strain and EMC strain. As we expected, this antibody showed potent neutralizing activity against both MERS-EMC and KNIH-002 strains.

## Binding affinity of mAbs to S1 and the RBD

The binding affinity of the selected mAbs to immobilized RBD or S1 protein on a CM5 chip was assessed by SPR assay. The $K_D$ (M) values for S1 ranged between 6.56 nM and <0.0753 nM, whereas those for the RBD ranged between 5.85 nM and 0.016 nM. Because the $K_d$ (1/s) values of KNIH77-A5 and KNIH77-B2 were above the device-detection limit ($10^{-5}$ s$^{-1}$), the results are presented using the inequality. KNIH77-A5 showed the highest affinity to both S1 and RBD proteins, whereas KNIH90-F1, which had the highest neutralizing ability, showed the next strongest affinity to RBD (Table 2).

## Computational structure modeling and the RBD and mAb interaction

The structures for the 240 amino acids of RBD and the variable region of KNIH90-F1 were predicted using I-TASSER and Antibody in the Rosetta online service. The RBD comprises seven α-helices (residues 385–9, 411–5, 430–5, 453–5, 463–8, 525–9, and 547–9) and 11 β-strands (residues 400–4, 407–9, 419–26, 438–47, 477–83, 497–507, 513–5, 538–44, 553–61, 568–76, and 583–6). The variable-region structure model of KNIH90-F1 was generated from residues 20 to 149 of the heavy chain and residues 20 to 129 of the light chain. A 14-amino acid peptide sequence of the complementarity determining region H3 loop for KNIH90-F1 was predicted at position 124SYGSGSYYTHYYAM137 of the heavy chain (Fig 3A). Modeled structures were used as the starting structures for docking simulation. According to the Snug-Dock protocol, these two structures were located very close to each other but did not super-pose. Determination of the RBD interaction with KNIH90-F1 was performed by the SnugDock algorithm based on computational modeling of KNIH90-F1 and RBD protein

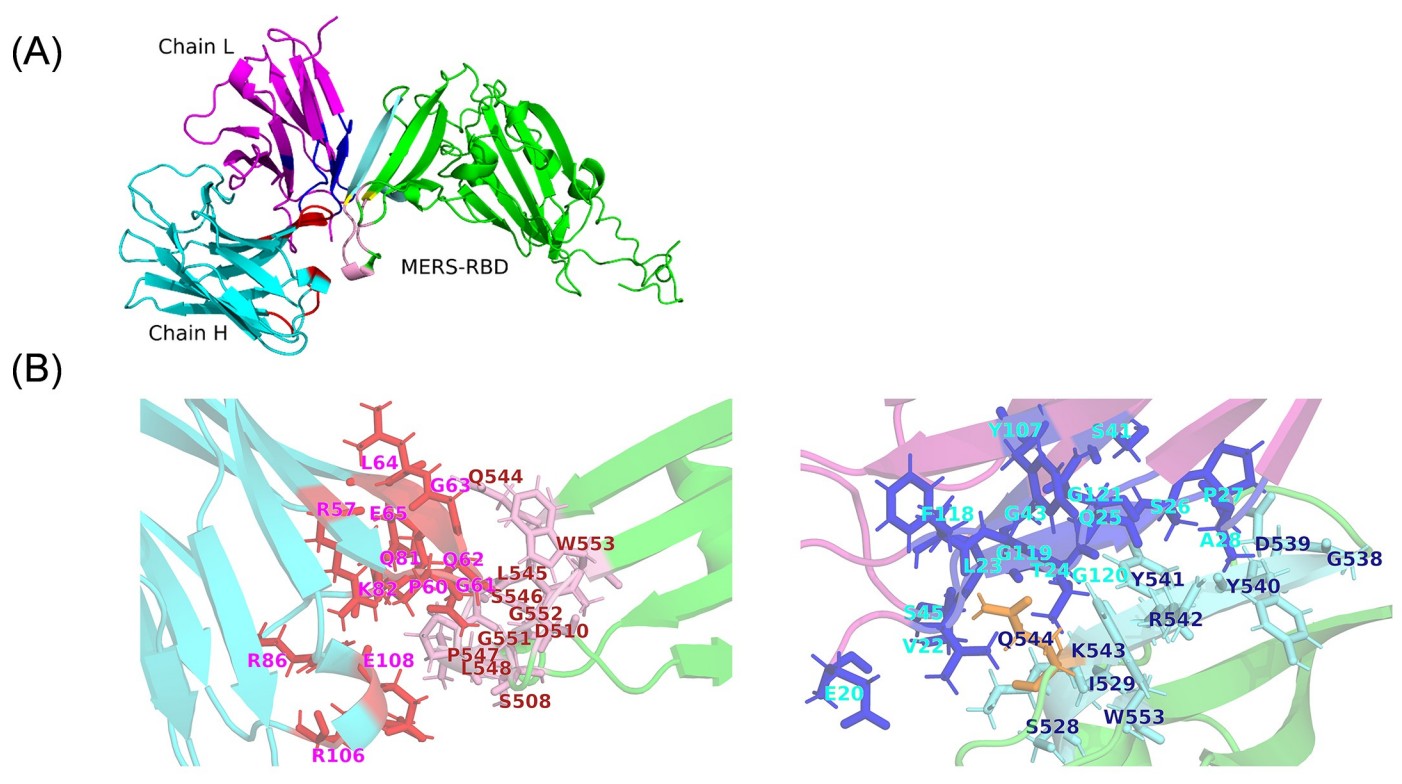

**Fig 3. Proposed docked structure of the RBD and mAb.** (A) A docked complex of the RBD and the KNIH90-F1 variable region. Green: RBD; cyan: heavy chain; magenta: light chain; red: interface residues of RBD with the heavy chain; pink: interface residues of the heavy chain; blue: interface residues of RBD with light chain (L); sky blue: interface residues of the light chain; yellow: overlapped interface residues in the heavy and light chains. (B) Interaction of the RBD and KNIH90-F1 complex. Interface residues of the RBD and heavy chain (left) and light chain (right). (C) Summarized putative interaction residues of the RBD, heavy chain, and light chain.

sequences (Fig 3B). SnugDock alignment indicated that the RBD N-terminus couples to the KNIH90-F1 variable region through four RBD-interfacing peptide sequences: 528SI529, 538GDYYRKQ544, 544QLSPL548, and 551GGW553. Another three individual amino acid sites (S508, D510, and W553) of the MERS-RBD molecule also aligned closely with the KNIH90-F1 variable region (Fig 3C). The RBD residues (510D, 538GDY540, and 553W) attached to the KNIH90-F1 antibody are the same sites that interact with DPP4 [35]. The several residues (508S, 510D, 538G, 543KQLS546, 548L, 551GG552) of binding interface in KNIH90-F1 mAb were expected as novel binding resides in comparison with those of other MERS-CoV antibodies [10, 36, 37]. Five of the simulated binding-interface residues (510D, 539D, 540Y, 542R, and 553W) showed strong hindrance of RBD–hDPP4-receptor binding in the mAb m336 study [12]. These results indicated that KNIH90-F1 was capable of neutralizing MERS-CoV entry by preventing RBD attachment to DPP4.

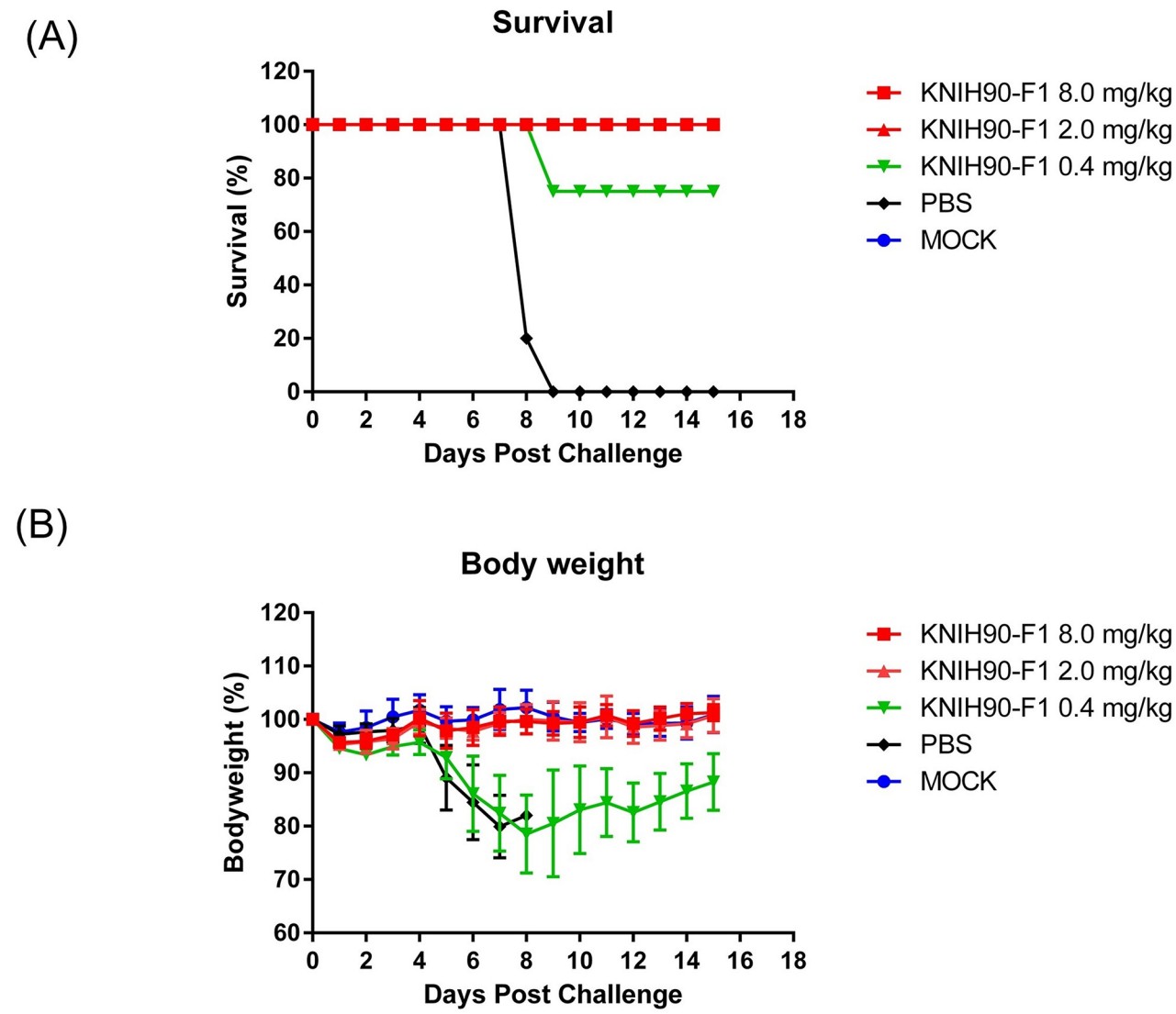

**Fig 4. MERS-CoV RBD-specific mAb protects mice from lethal challenge.** Five hDPP4 TG mice were intraperitoneally administered different doses of KNIH90-F1 24 h after intranasal inoculation with the MERS-CoV KNIH-002 strain ($1 \times 10^5$ pfu/30 μl). Body weight change (A) and survival (B) were monitored for 14 days.

### Efficacy of MERS-CoV mAbs in hDPP4 TG mice

The mAb KNIH90-F1 was selected for the animal study, because it exhibited the highest neutralizing potency *in vitro* assays. Human DPP4-expressing C57BL/6N TG mice were highly susceptible to MERS-CoV infection, demonstrating an analytical $LD_{50}$ of wild-type MERS-CoV KNIH-002 of $1 \times 10^3$ pfu (S2 Fig). To evaluate the therapeutic efficacy of KNIH90-F1, mice were challenged with a high dose (100 $LD_{50}$) of the EMC strain. After 24 h, the mice were intraperitoneally administered KNIH90-F1 at different concentrations. Change in body weight and survival of mice were monitored for 14 days. All placebo (PBS)-treated mice died within 9-days post-infection (Fig 4B); however, groups treated with 50 μg (2 mg/kg) and 200 μg (8 mg/kg) of the mAb were completely protected from the lethal challenge and exhibited only a minimal body weight change (Fig 4A). The mice in group that received the lowest dose of the

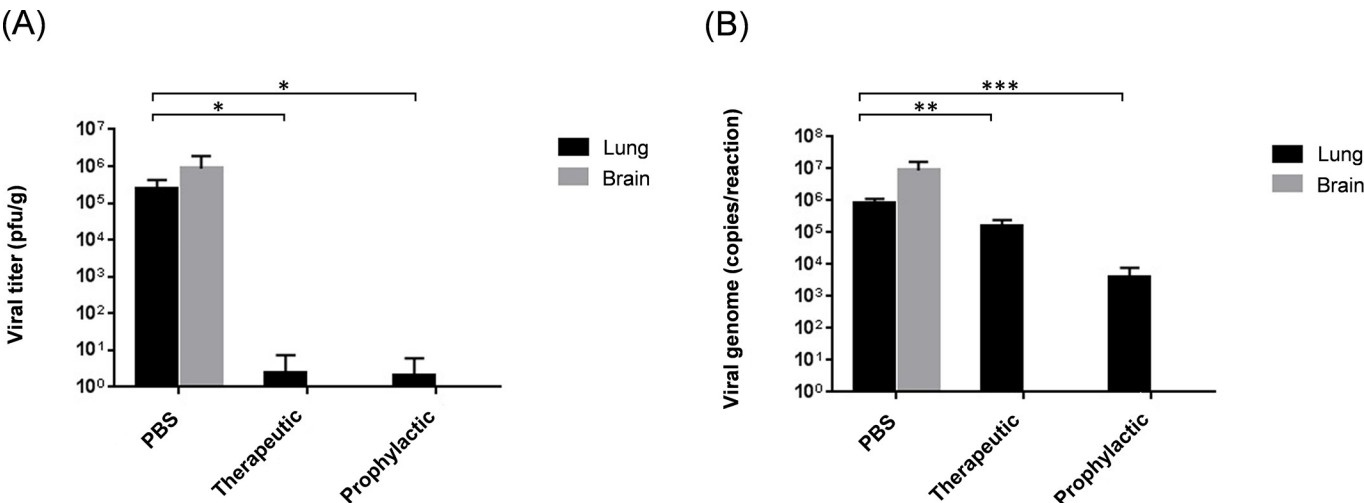

**Fig 5. MERS-CoV RBD-specific mAb inhibits viral replication in mouse tissues.** Four mice were treated with the KNIH90-F1 mAb (8 mg/kg) 24 h after MERS-CoV KNIH-002 strain lethal challenge ($1 \times 10^5$ pfu/mouse) under therapeutic conditions. Moreover, four mice were infected with MERS-CoV 24 h after administration of the same amount of mAb under prophylactic conditions. At 4-days post-challenge, all mice were humanely sacrificed, and tissues (lung and brain) were collected for viral titer analysis by plaque assays (A) and real-time RT-PCR (B). Viral genome copies in tissues were determined by real-time RT-PCR targeting the *Orf1a* gene. Data represent the mean ± standard deviation. $^*P \leq 0.05$; $^{**}P \leq 0.005$; $^{***}P \leq 0.0005$.

mAb (10 μg; 0.4 mg/kg) showed an 80% survival rate with significant weight loss and recovered after 8 days post-infection (Fig 4B).

To examine the viral loads of lung and brain tissues in both prophylactic and therapeutic settings, mice were sacrificed at 4-days post-infection. No infectious virus was detected in the brain for both the therapeutic and prophylactic administrations of the mAb. Similarly, infectious viruses were hardly detected in the lungs, with a ~5 log reduction as compared with the control group (Fig 5A). However, high numbers of copies of the viral genome were detected in the lungs, indicating that the residual viral genome or non-replicable viral particles persisted (Fig 5B). The previous report elucidated that the mice infected with MERS-CoV had shown the pathological changes, gross pathological changes, mononuclear cell infiltration and alveolar edema on day 4 and day 7 post-infection [13], we performed euthanasia on day 4 and 7 to analyze the therapeutic or prophylactic effect of mAbs. Histopathological changes, such as inflammatory cell infiltration, neurogenic necrosis, and perivascular inflammatory cell infiltration, were observed in lung and brain tissues of the untreated group, as described previously [13]. (Fig 6). Additionally, monocyte infiltration was observed in lung tissues of both KNIH90-F1-treated and PBS-treated groups at 4-days post-infection but not in brain tissues. Inflammatory monocyte infiltration was observed in the brains of PBS-treated mice 7-days post-infection, whereas mAb-treated mice showed no apparent brain lesions. These data suggest that MERS-CoV infection induced the influx of monocytes into the lungs and brains of infected mice in a time-dependent manner.

## Discussion

MERS-CoV cases have been reported continuously in the Middle East since the virus was first identified in April of 2012 [1]. The largest outbreak of MERS-CoV infections outside the Middle East occurred in Korea in 2015 and resulted in 186 laboratory confirmed cases with 38 deaths [3]. Since then, the importance of patient management, especially in hospitals, and preparedness against newly emerging infectious diseases has been emphasized from a public health perspective. Due to the absence of approved vaccines and virus-specific antiviral agents,

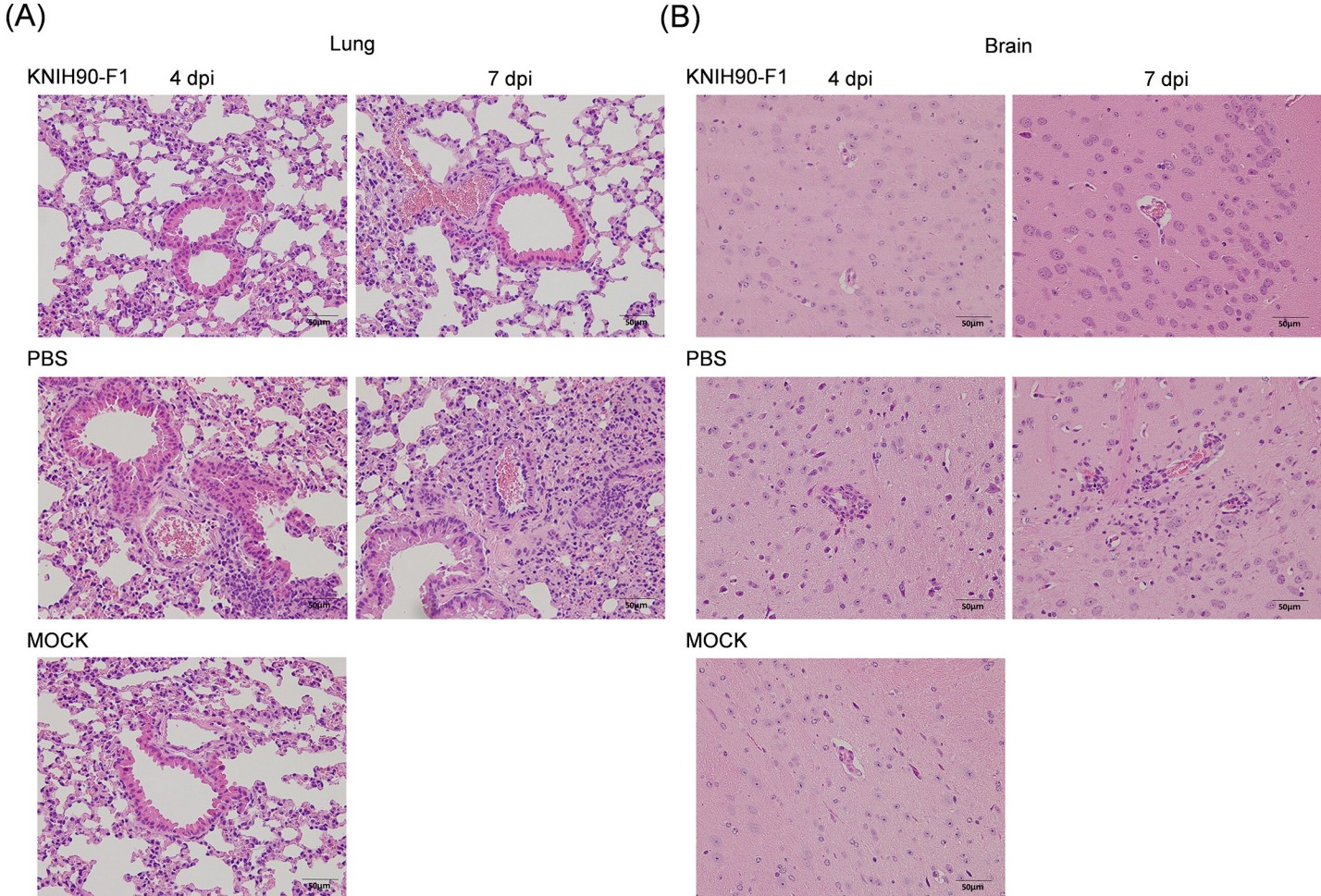

**Fig 6. Histopathological changes of the lungs and brains in MERS-CoV-infected mice.** Six mice were treated with the KNIH90-F1 mAb (8 mg/kg) 24 h after lethal challenge. At 4- and 7-days post-challenge, all mice were sacrificed, and lungs (A) and brains (B) were collected. Tissue sections were stained with H&E (magnification: 400×).

an alternative antiviral-treatment guideline for MERS-CoV patients involving the triple administration of type I interferon, ribavirin, and lopinavir/ritonavir for 10 to 14 days was recommended in South Korea [38]. However, side effects of this triple administration and significantly low therapeutic efficacy of ribavirin and interferon for severe MERS cases have been reported [39–42]. Therefore, a virus-specific neutralizing antibody regimen could be proposed as an alternative treatment.

In this study, single B cells were directly isolated from the PBMCs of convalescent MERS-CoV-infected Korean patients using flow cytometry cell sorter with a prefusion MERS-CoV S-2P probe and/or S1, and the IgG genes were promptly cloned for expression in mammalian cells. Previous reports, including MERS-CoV, SARS-CoV, HIV and RSV, showed that structure-designed stabilization of the viral surface protein by mutations or multivalent display induced greater neutralizing mAbs than postfusion proteins [20–23]. Accordingly, we reasoned that the prefusion S-2P probe could bind greater neutralizing antibodies on the surface of single B cells than wild S protein. Considering the duration from the isolation of single B cells to the purification of IgGs, we obtained total 11 mAbs rapidly within one month by using

this antibody technology. All these mAbs had high specificity, bound to only the cells infected with MERS-CoV, not other coronaviruses in the IFA experiment.

Among the six RBD-specific mAbs obtained, the most potent neutralizing antibody was KNIH90-F1, which showed exceptional therapeutic efficacy in the mouse model. In line with other studies, the RBD-specific antibodies showed high neutralizing activity, whereas the $IC_{50}$ values from the live-virus-based assay varied widely [6–10, 12, 43]. It is speculated that RBD-specific mAbs, such as LCA60 ($IC_{50}$ = 0.110–0.279 μg/ml), MERS-GD27 ($IC_{50}$ = 0.001 μg/ml) and KNIH90-F1 ($IC_{50}$ = 0.006 μg/ml), generated directly from patient B cells repertoires are more potent and are thus more likely to neutralize wild-type viruses. However, among the S1 (non-RBD) mAbs, only KNIH90-B2 and KNIH90-B7 showed mild neutralizing ability against the live virus (failed to reach the $IC_{50}$ value). The N-terminal domain (NTD) of S1 binds sialic acid of host cellular receptors [44], and non-RBD mAbs targeting the NTD of S1 also show neutralizing activity [5]. We assumed that KNIH90-B2 and KNIH90-B7 could bind to NTD of S1, resulting the mild neutralization via preventing the viral attachment to the sialic acid of cellular receptors. However, no S2-specific antibody was isolated from patient PBMCs in the present study. This might be attributed to the lower immunogenicity of the S2 than the S1 domain, as is the case for the influenza virus HA2 stalk domain [45]. The practical use of antibody cocktails has been used for Ebola virus [46, 47], hepatitis C virus [48], and MERS-CoV [17] to prevent the emergence of escape mutants as well as enhance the neutralization breadth. We suggest that combining RBD mAbs and non-RBD mAbs, targeting multiple antigenic sites on MERS-CoV spike, need to be implemented for treating future MERS-CoV infection.

As MERS-CoV does not naturally infect mice, we employed TG mice that systemically expressed the human receptor hDPP4 for the challenge experiment. All KNIH90-F1-treated mice survived after challenge with an extremely high dose (100 $LD_{50}$; $1 \times 10^5$ pfu/mouse) of the virus, including those that received the low dose (2 mg/kg) of mAbs, reflecting the potent therapeutic efficacy of this mAb. Additionally, the residual live-virus particles were not detected in the lungs by 4-days post-infection in both the therapeutic and prophylactic administration groups. Even with a significant amount of viral genome in the lungs, MERS-CoV infection rarely occurred through plaque assays. It was estimated that the KNIH90-F1 mAb administered to the animal model remained in the tissues or attached to the virus and neutralized its infectivity. Although there is substantial heterogeneity in drug disposition and pharmacokinetics of mAbs, the half-life of mAbs is generally 14 days (7–21 days) [49]. We sacrificed the mice administered with KNIH90-F1 mAb on the post-infection four days. Therefore, we assumed that a sufficient amount of mAb in the lungs could block the remained MERS-CoV infection in plaque assays.

Despite the promise of our developed mAbs, there are several limitations of this study that should be addressed in the future. First, escape mutants are more likely to occur when cultured viruses are continuously exposed to RBD-specific antibodies [10, 15]. Because KNIH90-F1 recognizes similar RBD domains, the efficacy of combined therapy with non-RBD S1 antibodies or RBD antibodies targeting different epitopes on the RBD should be evaluated to reduce the potential for escape mutations. In addition to direct receptor inhibition, indirect Fc-receptor-mediated immune responses, such as antibody dependent cellular cytotoxicity and complement-dependent cytotoxicity, should also be investigated. Second, due to the limited availability of TG mice, survival analysis following prophylactic administration was not completed. Only a pathogenic study that included measurements of viral titers in lung and brain tissues and histopathology were conducted. Therefore, additional animal experiments should be performed to assess complete prophylactic efficacy. Third, computer-assisted structural docking modeling suggested that the putative interacting residues on RBD are similar to those published previously [5, 8, 12]; however, more precise epitope mapping, such as that utilizing X-

ray crystallography or RBD point-mutation analysis, is needed to resolve the exact epitope structure.

In conclusion, we generated a highly potent human neutralizing mAb (KNIH90-F1) against MERS-CoV infection by using a prefusion-stabilized S-2P probe, and a direct IgG cloning platform from isolated B cells. This approach could be promptly applied for newly emerging viruses potentially threatening public health in the future. The KNIH90-F1 mAb could represent a potential candidate for development of a novel MERS therapeutic agent, as well as a prophylactic treatment. Moreover, it could be used in combination with other mAbs targeting different RBD epitopes or non-RBD mAbs for practical MERS immunotherapy.

## Supporting information

**S1 Fig. Neutralizing activity of KNIH90-F1.** Neutralization potency of the RBD-specific mAb KNIH90-F1 was evaluated by a PRNT assay using low mAb concentrations (0.00025–0.25 μg/ml). Percent inhibition by the mAb was determined by comparison with untreated cells (A). The $IC_{50}$ value was determined by nonlinear regression curve fitting (B). Each experiment was performed in triplicate, and data represent the mean of each experiment with standard errors.
(TIF)

**S2 Fig. Determination of 50% mouse lethal dose ($LD_{50}$).** (A) Body weight loss curve of 10-week-old male C57BL/6N-hDPP4 mice (n = 5 per group) infected with different doses of MERS-CoV KNIH-002. The error bar indicate the standard error of the mean. The weight loss and survival were monitored for 14 days.
(TIF)

**S3 Fig. Specificity of human monoclonal antibodies for MERS-CoV.** IFA was performed as described previously [27]. Briefly, Vero cells were infected with MERS-CoV Erasmus (EMC) strain and SARS-CoV-2 isolated from a Korean patient [50]. MRC5 cells were infected with human coronaviruses (hCoV) OC-43 strain (ATCC, Manassas, VA, USA) and 229E strain (ATCC). LLC-MK2 cells were infected with hCoV NL63 strain (CN061/14). After the cytopathogenic effect appeared in the infected cells, the cells were fixed with 70% methanol in PBS. Then, human anti-MERS-CoV antibodies were incubated with the Vero cells infected with MERS-CoV. As positive controls, a rabbit polyclonal anti-MERS-CoV spike protein antibody (Sino biological Inc., Beijing, China) for MERS-CoV, a mouse anti-229E coronavirus nucleoprotein OC-43 antibody (MERCK, Darmstadt, Germany) for HCoV-229E, a mouse anti-coronavirus antibody, hCoV OC-43 (LifeSpan BioScience, Seattle, WA, USA) for HCoV-OC43, a rabbit polyclonal anti-hCoV-HKU1 spike protein antibody (Sino Biological Inc., Beijing, China) for hCoV-NL63, a human serum of a Korean SARS-CoV-2 convalescent person were also incubated. The IFA performed without the positive primary antibodies (described above) were used as negative controls. The secondary antibodies, Fluorescein isothiocyanate (FITC) -conjugated goat anti-human-IgG (Abcam, Cambridge, U.K.), FITC-conjugated goat anti-rabbit-IgG (Abcam, Cambridge, U.K.), and FITC-conjugated goat anti-mouse-IgG (Jackson Immunoresearch, West Grove, PA, USA) were added to the cells and incubated. The cells were observed, and images were obtained using a fluorescence microscope.
(TIF)

## Acknowledgments

We thank members of the New Drug Development Center, Osong Medical Innovation Foundation for the insightful analysis of the binding affinity of mAbs. We also thank Wan Beom

Park (Seoul National University College of Medicine) for the kind supply of SARS-CoV-2 convalescent serum. We are grateful to Jun Won Kim for support IFA with SARS-CoV-2 virus.

## Author Contributions

**Conceptualization:** Lingshu Wang, Barney S. Graham, John R. Mascola, Nanshuang Wang, Jason S. McLellan, Hansaem Lee.

**Data curation:** Jang-Hoon Choi, Hansaem Lee.

**Formal analysis:** Sang-Mu Shim.

**Funding acquisition:** Sung Soon Kim, Joo-Yeon Lee.

**Investigation:** Sung Soon Kim, Barney S. Graham, John R. Mascola, Joo-Yeon Lee.

**Methodology:** Jang-Hoon Choi, Hye-Min Woo, Tae-young Lee, So-young Lee, Woo-Jung Park, Joo Ae Kim, Yi Zhang, Wei Shi, Lingshu Wang.

**Project administration:** Sung Soon Kim, Joo-Yeon Lee, Hansaem Lee.

**Resources:** Wei Shi, Nanshuang Wang, Jason S. McLellan.

**Software:** Mi-Ran Yun, Dae-Won Kim.

**Supervision:** Sung Soon Kim, Barney S. Graham, Hansaem Lee.

**Validation:** Jang-Hoon Choi, So-young Lee, Jeong-Sun Yang, Barney S. Graham, Hansaem Lee.

**Visualization:** Mi-Ran Yun.

**Writing – original draft:** Jang-Hoon Choi, Hansaem Lee.

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
