## [Decision Letter · Decision Letter 0]

29 Jan 2020

PONE-D-19-34811

Characterization of a human monoclonal antibody generated from a B cell specific for a prefusion-stabilized spike protein of 

Middle East respiratory syndrome coronavirus

PLOS ONE

Dear PhD Lee,

Thank you for submitting your manuscript to PLOS ONE. It has been reviewed by three experts in the field. All reviewers felt that your study has merit but also made important suggestions for improvement. Therefore, I invite you to submit a revised version of the manuscript that addresses the points raised during the review process.

I would appreciate receiving your revised manuscript by Mar 14 2020 11:59PM. To enhance the reproducibility of your results, we recommend that if applicable you deposit your laboratory protocols in protocols.io, where a protocol can be assigned its own identifier (DOI) such that it can be cited independently in the future. For instructions see: http://journals.plos.org/plosone/s/submission-guidelines#loc-laboratory-protocols

I look forward to receiving your revised manuscript.

Kind regards,

Stefan Pöhlmann, Ph.D.

Academic Editor

PLOS ONE

Journal Requirements:

2. In your Methods section, please give the source of the animals used in your study.

3. We noticed you have some minor occurrence(s) of overlapping text with the following previous publication(s), which needs to be addressed:

https://doi.org/10.3390/antib8030042

In your revision ensure you cite all your sources (including your own works), and quote or rephrase any duplicated text outside the Methods section. Further consideration is dependent on these concerns being addressed.

4. In the ethics statement in the manuscript and in the online submission form, please provide additional information about the patient records/samples used in your retrospective study.

Specifically, please ensure that you have discussed whether all data/samples were fully anonymized before you accessed them and/or whether the IRB or ethics committee waived the requirement for informed consent.

If patients provided informed written consent to have data/samples from their medical records used in research, please include this information.

5. Please note that all PLOS journals ask authors to adhere to our policies for sharing of data and materials: https://journals.plos.org/plosone/s/data-availability. According to PLOS ONE’s Data Availability policy, we require that the minimal dataset underlying results reported in the submission must be made immediately and freely available at the time of publication. As such, please remove any instances of 'unpublished data' or 'data not shown' in your manuscript and replace these with either the relevant data (in the form of additional figures, tables or descriptive text, as appropriate), a citation to where the data can be found, or remove altogether any statements supported by data not presented in the manuscript.

6. We note that you have a patent relating to material pertinent to this article.

a. Please provide an amended statement of Competing Interests to declare this patent (with details including name and number), along with any other relevant declarations relating to employment, consultancy, patents, products in development or modified products etc.

Please confirm that this does not alter your adherence to all PLOS ONE policies on sharing data and materials, as detailed online in our guide for authors http://journals.plos.org/plosone/s/competing-interests by including the following statement: "This does not alter our adherence to  PLOS ONE policies on sharing data and materials.” If there are restrictions on sharing of data and/or materials, please state these.

Please note that we cannot proceed with consideration of your article until this information has been declared.

7. Please amend the manuscript submission data (via Edit Submission) to include author Wei Shi.

Reviewers' comments:

Reviewer's Responses to Questions

**Comments to the Author**

1. Is the manuscript technically sound, and do the data support the conclusions?

Reviewer #1: Partly

Reviewer #2: Partly

Reviewer #3: Partly

2. Has the statistical analysis been performed appropriately and rigorously? 

Reviewer #1: No

Reviewer #2: N/A

Reviewer #3: Yes

3. Have the authors made all data underlying the findings in their manuscript fully available?

Reviewer #1: No

Reviewer #2: Yes

Reviewer #3: Yes

4. Is the manuscript presented in an intelligible fashion and written in standard English?

Reviewer #1: Yes

Reviewer #2: Yes

Reviewer #3: Yes

5. Review Comments to the Author

Reviewer #1: This study describes the isolation and recombinant production of a set of MERS CoV-specific IgG, in vitro functional testing of specific clones and subsequent testing of a single clone for its therapeutic potential in an experimental infection mouse model. The manuscript is well written although type-editing is required to improve readability. The data supports the overall conclusion that the tested clone is a potential candidate as a prophylactic or therapeutic antiviral but more detailed and larger studies are warranted to substantiate this claim. Several points need to be addressed to corroborate and clarify their findings as listed below.

General comments:

The rationale for choosing S-2P as a bait for isolation of B cells is not explained clearly or emphasized enough.

Only a single clone is tested for its therapeutic potential in vivo, despite the functional differences observed in vitro. Therefore, the impact of in vitro differences is not assessed. Other monoclonal antibodies with therapeutic potential from this study or another publication for a complete evaluation of KNIH90-F1 potential would be of added value.

These data should be confirmed by mutations in recombinant protein and assessment of binding, or generation of neutralization escape mutants.

The statistics main section, as well as its description at the figure legends, is too simplified and should be more accurately described (correctly referencing replicates per experiment and the test that was performed).

Specific comments:

• Line 49: please update since this reference is almost 1 year old.

• Line 57: neutralizing antibodies do not bind the host receptor, but instead bind the viral RBD to prevent its binding to the receptor.

• Line 63, please remove “Chinese”and “Korean” from the main text, as this detail is already in the reference.

• Line 71: you only reference results from 1 clinical trial, what about the humanized mAbs (REGN)?

• Line 106: specify the gravitational force at 1500 rpm

• Line 110: the authors need to provide more information about the gating strategy and sorting of S-2P and S1 specific B cells. An explanation for choosing the specific mix of antibodies is missing.

• Line 113: the usage of S1 in combination with fluorescently labeled S-2P for B cell sorting is not explained.

• Line 119: heading needs to be more specific

• Line 141: a more detailed explanation of the evaluation of the antibody sequences using IMGT-QUEST tool is missing.

• Line 152: the reason why S2 fused with murine Fc is used for ELISA is not specified.

• Line 201: please replace “C196ages” with Cages.

• Line 216, 224, 229: day 4 and day 7 post-infection euthanasia is not argued.

• Line 217: it is not specified why only brain and lung tissues were selected for pathology studies. The authors do not specify the weight of tissue that was analyzed in each case, which is necessary for comparison of titers as titers are represented as pfu/g.

• Line 218. Please provide details on how the plaque assay was performed.

• Line 226: First line should be moved to previous section.

• Line 233: Please specify how the number of genome copies is determined.

• Line 248: PBMCs of five different convalescent MERS patients were isolated, but only three were used. Nothing is mentioned about the other two patients or why these 3 were selected.

• Line 252: it is not specified how these 90 B cells were selected for sorting, and which patients’ they were isolated from. Furthermore, sequences of variable heavy chain and kappa or lambda light chain regions should be submitted to an online database, such as GenBank, for future reference.

• Line 259: the meaning of this sentence is unclear. An S2 antibody was not identified although this domain is relatively conserved as compared to S1?

• Line 285: Compared to what strain, EMC?

• Line 291, Table 1: the number of replicates performed on this experiment is not specified.

• Line 334: These data should be confirmed by mutations in recombinant protein and assessment of binding, or generation of neutralization escape mutants.

• Line 365: What is the rational for selectively examining lung and brain tissues? Longitudinal analysis of swabs and peripheral blood would be of added value.

• Line 366: what is the rationale for selecting day 4?

• Line 369: or more likely, as indicated in the discussion, the presence of neutralizing antibody will neutralize any infectious virus when titrating.

• Line 374: on what data do the authors base the “time-dependent manner”?

• Line 374: infiltration of monocytes should be quantified in order to determine differences.

• Line 414: the term “wild S-probe” is incorrect.

• Line 422: the meaning of this sentence is not clear. No data is available that allows speculation on how “intact” the repertoire of B-cells using different methods to isolate specific clones. What is meant by “intact”? does this refer to how representative the B-cell repertoire is?

• Line 441: “superior therapeutic efficacy” can only be claimed when testing the antibodies side-by-side in the same model.

• Line 448: the explanation on the choice of post-infection day for euthanasia could be more detailed (and maybe placed in a different section).

• Line 453: evaluation of neutralization activity of KNIH90-F1 should be already evaluated on escape mutants.

Figure 1: Technical or experimental replicates are not specified.

Figure 2A: Lack of error bars suggests a single measurement is performed, yet legend suggests triplicates. Please clarify. Title on right graph has a type error.

Figure 3B: It is not clear on what protein each aa is located in this graphical representation. The use of for instance colored fonts may help to clarify.

Reviewer #2: Currently there are no approved therapies for MERS. Many groups have focused on the identification and characterization of antibodies against MERS-CoV. Now Choi et al., analysed a human monoclonal antibody generated from a B cell specific for a prefusion-stabilized spike protein of Middle East respiratory syndrome coronavirus. Overall, the methods used are sound and the manuscript is well written. However, I have one major issue with this manuscript. The authors use the prefusion-stabilized spike protein and claim that the antibodies are specific for the prefusion form. This is not the case. In fact this has not been demonstrated in the study, there is no specific interaction with this prefusion form, the antibodies recognize also the S1 protein they use...

The confusion to me starts in the methods section where they indicate that B cells are sorted that are positive for the spike prefusion protein AND S1. So the authors should comment on that in detail because there is no specific selection of prefusion antibodies. If this indeed is the case the manuscript should be rewritten to take out this claim.

Furthermore, antibodies to S1 that do not recognize RBD are not well characterized. The suggest these recognize the A domain, but this has not been analyse. Studies to show the antigen specificity need to be provided.

Reviewer #3: Re: PONE-D-19-34811

The manuscript, entitled “Characterization of a human monoclonal antibody generated from a B cell specific for a prefusion-stabilized spike protein of Middle East respiratory syndrome coronavirus” by Jang-Hoon Choi, et al, describes novel human monoclonal antibodies specific to MERS-coronavirus spike protein developed by cell-sorting of B cells obtained from MERS-recovered patients. The RBD-recognizing antibodies exhibit a high neutralization potency against MERS-CoV in vitro. Among them, one MAb (KNIH90-F1) is effective for prophylaxis or treatment of MERS-CoV infection in mice model. Overall the presented data are convincing. Although the development of therapeutic candidates of anti-MERS-CoV MAb is very important, a major concern is the lack of novel contribution to research on anti-MERS-CoV MAb except for B-cell sorting methodology. As authors have already mentioned, there are already several reports describing MAbs with therapeutic and prophylactic potential against MERS-CoV infection.

Several points that need to be described in the text.

1. It is not clear what is concluded from the data showing MAb binding affinity (Table 2) and 3D structure modeling (Fig.3). Most of the data can be omitted.

2. Line 80 or 469, how can you say “fastest” or “rapidly and conveniently”?

3. Line 143, an information of expression vector (plasmid name, company, etc) is missing.

4. Line 165, please indicate the details of the pseudovirus. Is this express luciferase, GFP, or others? Also, this information should be included in figure 2.

5. Line 261, “as this domain is relatively conserved…” seems to be incorrect. S1 positive clones were selected in the B cell sorting as described in materials and methods (line 113). The same thing seems to be applied to line 429 “no S2-specific antibody was isolated…”.

6. Line 283, table 1 does not indicate the data of “no difference between IC50 values of MERS-CoV EMC and KNIH-002..”.

7. Line 285, “change” should be “differences”.

6. PLOS authors have the option to publish the peer review history of their article (what does this mean?). If published, this will include your full peer review and any attached files.

Reviewer #1: No

Reviewer #2: No

Reviewer #3: No

---

## [Author Response · Author response to Decision Letter 0]

1 Apr 2020

Reviewer #1: This study describes the isolation and recombinant production of a set of MERS CoV-specific IgG, in vitro functional testing of specific clones and subsequent testing of a single clone for its therapeutic potential in an experimental infection mouse model. The manuscript is well written although type-editing is required to improve readability. The data supports the overall conclusion that the tested clone is a potential candidate as a prophylactic or therapeutic antiviral but more detailed and larger studies are warranted to substantiate this claim. Several points need to be addressed to corroborate and clarify their findings as listed below.

General comments:

The rationale for choosing S-2P as a bait for isolation of B cells is not explained clearly or emphasized enough.

☞ We added the explanation of reference 20 in Introduction to explain why we chose S-2P as a bait. At first, S-2P was expressed by 20-fold dramatically as well as it induced more neutralizing antibodies in mice, compared to wild S, as described previously such as SARS-CoV, HIV, and RSV (reference 20, 21, 22, and 23) in line 84-86. Thus, we hypothesized that structure-designed stabilized viral surface proteins could bind the neutralizing antibodies of convalescent patients’ B cells more efficiently than wild-type protein.

Only a single clone is tested for its therapeutic potential in vivo, despite the functional differences observed in vitro. Therefore, the impact of in vitro differences is not assessed. Other monoclonal antibodies with therapeutic potential from this study or another publication for a complete evaluation of KNIH90-F1 potential would be of added value.

☞ We agreed with your opinion. At first, we have been testing the synergic effect of antibody cocktails in vitro assays. Then, we will try the combination test for the therapeutic efficacy in animal model. However, it requires lots of time, labors, and transgenic mice. The minimum two months are needed to produce the amount of hDPP4 transgenic mice sufficient for the animal tests. We will show the further result later. 

These data should be confirmed by mutations in recombinant protein and assessment of binding, or generation of neutralization escape mutants.

☞ We agreed with your opinion. We have processed several immune escape mutants in the case of KNIH90-F1. We are also going to test pseudoviruses having Spike protein mutants to confirm whether the mutation site may be critical to the neutralizing antibodies. We will prepare this result for another publication. 

The main statistics section, as well as its description at the figure legends, is too simplified and should be more accurately described (correctly referencing replicates per experiment and the test that was performed).

☞ According to your advice, we added more explanation to the statistics at the figure legends.

Specific comments:

? Line 49: please update since this reference is almost 1 year old.

☞ We update the reference, 2,519 laboratory-confirmed patients and 866 deaths (34.3 % mortality) have been reported as of January 2020 in line 49-50.

? Line 57: neutralizing antibodies do not bind the host receptor, but instead bind the viral RBD to prevent its binding to the receptor.

☞ We correct the sentence, “Several laboratory studies proposed vaccine candidates based on the receptor-binding domain (RBD) of spike (S) antigen, which induce neutralizing antibodies that inhibit RBD from binding the host receptor dipeptidyl peptidase 4 (hDPP4; also known as CD26) [4]” in line 56-69.

? Line 63, please remove “Chinese”and “Korean” from the main text, as this detail is already in the reference

☞ We removed the words “Chinese” and “Korean” in line 64-65.

? Line 71: you only reference results from 1 clinical trial, what about the humanized mAbs (REGN)?

☞ We try to update the result of the clinical trial for humanized mAbs (REGN3048 and REGN3051); however, we could not get any result from ClinicalTrials.gov, pubmed, Google and the website of Regeneron, except the result of marmosets (Antiviral Res. 2018 Aug;156:64-71.) 

? Line 106: specify the gravitational force at 1500 rpm

☞ We corrected 1,500 rpm to 200 g force in line 110.

? Line 110: the authors need to provide more information about the gating strategy and sorting of S-2P and S1 specific B cells. An explanation for choosing the specific mix of antibodies is missing.

☞ As we mentioned before, we added the explanation of reference 20 (Pallesen J, Wang N, Corbett KS, Wrapp D, Kirchdoerfer RN, Turner HL, et al. Immunogenicity and structures of a rationally designed prefusion MERS-CoV spike antigen. Proc Natl Acad Sci U S A. 2017;114(35):E7348-e57.) in Introduction to explain why we chose S-2P as a bait. At first, S-2P was expressed by 20-fold dramatically as well as it induced more neutralizing antibodies in mice, compared to wild S, as described previously such as SARS-CoV, HIV, and RSV (reference 20, 21, 22, and 23). We hypothesized that structure-designed stabilized viral surface proteins could bind the neutralizing antibodies of convalescent patients’ B cells more efficiently than wild-type proteins.

? Line 113: the usage of S1 in combination with fluorescently labeled S-2P for B cell sorting is not explained.

☞ We described two probes usage in the line 118.

? Line 119: heading needs to be more specific

☞ We corrected the heading in line 125.

? Line 141: a more detailed explanation of the evaluation of the antibody sequences using IMGT-QUEST tool is missing.

☞ We added more explanation in the sentence in line 146-148.

? Line 152: the reason why S2 fused with murine Fc is used for ELISA is not specified.

☞ We added more explanation in the sentence since murine Fc was tagged to S2 to improve in vitro solubility and stability when expressed in the cells in line 160-161.

? Line 201: please replace “C196ages” with Cages.

☞ We corrected it in line 209.

? Line 216, 224, 229: day 4 and day 7 post-infection euthanasia is not argued.

☞ According to the reference 13 (Stalin Raj V, Okba NMA, Gutierrez-Alvarez J, Drabek D, van Dieren B, Widagdo W, et al. Chimeric camel/human heavy-chain antibodies protect against MERS-CoV infection. Sci Adv. 2018;4(8):eaas9667.), the mice infected with MERS-CoV showed the pathological changes, gross pathological changes, mononuclear cell infiltration and alveolar edema on day 4 and day 7 post-infection, we performed euthanasia on both time to analyze the therapeutic effect of mAbs. We added the explanation in the result text.

? Line 217: it is not specified why only brain and lung tissues were selected for pathology studies. The authors do not specify the weight of tissue that was analyzed in each case, which is necessary for comparison of titers as titers are represented as pfu/g.

☞ According to the reference 31 (Agrawal AS, Garron T, Tao X, Peng BH, Wakamiya M, Chan TS, et al. Generation of a transgenic mouse model of Middle East respiratory syndrome coronavirus infection and disease. Journal of virology. 2015;89(7):3659-70.), high infectious virus titers were detected from the lungs and brains of mice on day 4 post-infection, whereas viral RNAs were also detected in the heart, spleen, and intestine, indicating a disseminating viral infection.

Moreover, infected transgenic mice were able to activate genes encoding for many antiviral and inflammatory mediators within the lungs and brains, coinciding with the high levels of viral replication. Therefore, we determined to measure viral titers in the lungs and brains. We also measured the brain and lung tissue weights after we harvested the tissues. We added the phrase “After brain and lung tissue weights were measured” (line 227-228).

? Line 218. Please provide details on how the plaque assay was performed.

☞ We added the protocol in Methods of “Assessment of therapeutic and prophylactic effects of mAb in mice.”

To measure the virus titration, tissues were homogenized in DMEM using a manual homogenizer and centrifuged for 10 min. Viral RNAs were extracted from the supernatant by Maxwell RSC Viral TNA kit (Promega). They were assayed by using PowerChekTM MERS (upE & ORF1a) Real-time RT-PCR kit (Kogene biotech) according to the manufacturer's instructions. Vero cell monolayers were inoculated with serial 10-fold dilutions of the supernatant, which was adsorbed for 1 hr. The cells were covered with 1 ml or 1.5% CMC solution. After 3 days, the cells were washed with PBS and fixed with 4% formaldehyde and stained with crystal violet.

? Line 226: First line should be moved to previous section. 

☞ We found that the same experiment was described in the “Assessment of therapeutic and prophylactic effects of mAb in mice” and “Histology”. Thus, we corrected the first line to “Six mice in each group were performed according to the same experiment design for the therapeutic efficacy in the previous section” to avoid the sentence duplication according to your advice (line 258-259).

? Line 233: Please specify how the number of genome copies is determined.

☞ We added the method “Assessment of therapeutic and prophylactic effects of mAb in mice” to specify how to construct standard RNAs and to calculate the number of genome copies. Briefly, upE of MERS-CoV RT-PCR was performed with SuperScript III reverse transcriptase (Invitrogen, Carlsbad, CA, USA) using MERS-CoV EMC as an RNA template. For PCR, 5 μl of the RT reaction was added to 20 μl of PCR amplification master mix containing Ex-Taq™ polymerase (Takara, Shuzo, Japan). After PCR purification with a PCR purification kit (Qiagen), 1 μl of pCR®2.1-TOPO® vector (Invitrogen) was added to the purified PCR products, and the mixture was transformed into One Shot® Top10 competent cells (Invitrogen). The standard RNAs were synthesized massively using a MEGAscript® T7 kit (Applied Biosystems) and the template plasmids pCR2.1-upE. The RNAs were purified by MEGAclear™ kit (Applied Biosystems). RNA copy numbers were calculated based on RNA concentration and size. The upE RNA fragment was diluted serially with a 0.5 log-dilution to yield 1 to 105 copies per reaction, and these dilutions were then used as templates for rRT-PCR. We calculated the Ct values of tissues and copy numbers. 

? Line 248: PBMCs of five different convalescent MERS patients were isolated, but only three were used. Nothing is mentioned about the other two patients or why these 3 were selected.

☞ Initially we prepared PBMCs from 5 patients but used those from only 3 patients therefore we corrected it (line 279).

? Line 252: it is not specified how these 90 B cells were selected for sorting, and which patients’ they were isolated from. Furthermore, sequences of variable heavy chain and kappa or lambda light chain regions should be submitted to an online database, such as GenBank, for future reference.

☞ We added more explanation in the sentence to describe how to sort B cells. The amino acid sequences were reported by the patent (“MONOCLONAL ANTIBODY FOR SPIKE PROTEIN OF MIDDLE EAST RESPIRATORY SYNDROME CORONAVIRUS, AND USE THEREOF” and No. 10-1895229 (Korean patent) (PCT application No. PCT KR2018009754)). Also, the gene sequences will be submitted to GenBank.

? Line 259: the meaning of this sentence is unclear. An S2 antibody was not identified although this domain is relatively conserved as compared to S1?

☞ We agreed with you, we corrected the sentence not to make confusion. S2 domain is more conserved than S1 but we did not get any S2 specific antibody. Therefore, it seems to be less immunogenicity (line 290-292). 

? Line 285: Compared to what strain, EMC?

☞ You’re right, we added “compared to EMC stain.”

? Line 291, Table 1: the number of replicates performed on this experiment is not specified.

☞ We agreed with you, so we added the table description more, “ The data are expressed as mean IC50 (n=3).” Additionally, we corrected sentence materials and methods (line 180).

? Line 334: These data should be confirmed by mutations in recombinant protein and assessment of binding, or generation of neutralization escape mutants.

☞ We agreed with your opinion, we also have thought that further studies were needed. We have found that several escape mutants of MERS-CoV from KNIH90-F1, KNIH90-B2, and KNIH90-F2. We are also analyzing X-ray crystallography of KNIH90-F1 and RBD. To gather all the data will require some time. We are going to show this result for another publication.

? Line 365: What is the rational for selectively examining lung and brain tissues? Longitudinal analysis of swabs and peripheral blood would be of added value.

☞ According to reference 13 and 31, high infectious virus titers and histopathology were detected in the lungs and brains of mice, thus we determined to examine lung and brain tissues for the therapeutic efficacy in vivo. However, we are also trying to analyze the tissue distribution of MERS-CoV, including swabs and peripheral blood, in our transgenic mice. This result will be published in detailed.

? Line 366: what is the rationale for selecting day 4?

☞ As we mentioned previously, according to reference 13 and 31, high infectious virus titers were detected from the lungs and brains of mice on day 4 post-infection, thus we sacrificed the mice on 4 days post-infection. 

? Line 369: or more likely, as indicated in the discussion, the presence of neutralizing antibody will neutralize any infectious virus when titrating.

☞ We are the same as your comments. The presence of KNIH90-F1 in the blood will neutralize the MERS-CoV when titrating.

? Line 374: on what data do the authors base the “time-dependent manner”? 

? Line 374: The infiltration of monocytes should be quantified in order to determine differences.

☞ The sentence has been rearranged to clarify (line 416-417). Unfortunately, we did not quantify infiltration of monocytes. However, pathologic analysis was performed by an authorized histopathological laboratory (ChemOn Inc., Suwon, South Korea). To help understand the histopathological information, we attached the pathological score table of the lungs and brains from ChemOn Inc. They scored pathology in tissue slides according to Pristima Glossary and INHAND of the book “Standardized System for Nomenclature and Diagnostic Criteria ? Guides for Toxicologic Pathology (Xybion)”. (Score: from minimal (+1) to massive pathology (+5), Number examined of each group=3) 

According their report, the pathological change and infiltration of mononuclear cells in lungs and brains after MERS-CoV infection had been processed more severely on 7 days post-infection than 4 days post-infection. Therefore, we thought the pathological change, such as the influx of monocyte into the lungs and brains, become more serious in a time-dependent manner.

(You can see the table in the word file)

? Line 414: the term “wild S-probe” is incorrect.

☞ We corrected it to “wild S protein” (line 454).

? Line 422: the meaning of this sentence is not clear. No data is available that allows speculation on how “intact” the repertoire of B-cells using different methods to isolate specific clones. What is meant by “intact”? does this refer to how representative the B-cell repertoire is?

☞ We agree with your comment and corrected the sentence line 464.

? Line 441: “superior therapeutic efficacy” can only be claimed when testing the antibodies side-by-side in the same model.

☞ According to your comments, we deleted the previous reports and then, we correct “superior” to “potent” and deleted the comparison (line 481-482). 

? Line 448: the explanation on the choice of post-infection day for euthanasia could be more detailed (and maybe placed in a different section).

☞ As we mentioned previously, we chose post-infection 4 days and 7 days according to reference 13 and 31.

? Line 453: evaluation of the neutralization activity of KNIH90-F1 should be already evaluated on escape mutants.

☞ As we mentioned previously, we have processed several immune escape mutants for the evaluation of the neutralization of KNIH90-F1. 

? Figure 1: Technical or experimental replicates are not specified.

☞ We added the sentence, “Data points represent the mean of three technical replicates with standard errors” in figure 1 legend.

? Figure 2A: Lack of error bars suggests a single measurement is performed, yet legend suggests triplicates. Please clarify. Title on right graph has a type error.

☞ We added the error bars on figure 2.

Figure 3B: It is not clear on what protein each aa is located in this graphical representation. The use of for instance colored fonts may help to clarify.

☞ We change the colors of amino acids to clarify Figure 3B.

Reviewer #2: Currently there are no approved therapies for MERS. Many groups have focused on the identification and characterization of antibodies against MERS-CoV. Now Choi et al., analysed a human monoclonal antibody generated from a B cell specific for a prefusion-stabilized spike protein of Middle East respiratory syndrome coronavirus. Overall, the methods used are sound and the manuscript is well written. However, I have one major issue with this manuscript. The authors use the prefusion-stabilized spike protein and claim that the antibodies are specific for the prefusion form. This is not the case. In fact this has not been demonstrated in the study, there is no specific interaction with this prefusion form, the antibodies recognize also the S1 protein they use...

The confusion to me starts in the methods section where they indicate that B cells are sorted that are positive for the spike prefusion protein AND S1. So the authors should comment on that in detail because there is no specific selection of prefusion antibodies. If this indeed is the case the manuscript should be rewritten to take out this claim.

☞ In this study, we used prefusion S trimer and S1 for B cell selection. We collected all prefusion S trimer and/or S1 positive B cells and tested antibodies specificity by ELISA with S trimer, S1, RBD, S2 protein. Although these also bind to S1 and RBD, there is no direct evidence for the prefusion trimer specific antibody. Only KNIH90-C4 bound to prefusion S trimer more strongly than S1 protein in ELISA (Figure 1B) in line 292. Therefore we revised the sentence in line 509 and changed methods section (line 118).

Furthermore, antibodies to S1 that do not recognize RBD are not well characterized. The suggest these recognize the A domain, but this has not been analyse. Studies to show the antigen specificity need to be provided.

☞ We agree with your comment. We only tested S1 (non-RBD) antibodies with ELISA so we can’t describe specificity of those antibodies. That is the one of this study limitation and will be resolved by epitope mapping. Therefore, we just assumed possibility of those antibody specificity in line 482-483.

Reviewer #3: Re: PONE-D-19-34811

The manuscript, entitled “Characterization of a human monoclonal antibody generated from a B cell specific for a prefusion-stabilized spike protein of Middle East respiratory syndrome coronavirus” by Jang-Hoon Choi, et al, describes novel human monoclonal antibodies specific to MERS-coronavirus spike protein developed by cell-sorting of B cells obtained from MERS-recovered patients. The RBD-recognizing antibodies exhibit a high neutralization potency against MERS-CoV in vitro. Among them, one MAb (KNIH90-F1) is effective for prophylaxis or treatment of MERS-CoV infection in mice model. Overall the presented data are convincing. Although the development of therapeutic candidates of anti-MERS-CoV MAb is very important, a major concern is the lack of novel contribution to research on anti-MERS-CoV MAb except for B-cell sorting methodology. As authors have already mentioned, there are already several reports describing MAbs with therapeutic and prophylactic potential against MERS-CoV infection.

Several points that need to be described in the text.

1. It is not clear what is concluded from the data showing MAb binding affinity (Table 2) and 3D structure modeling (Fig.3). Most of the data can be omitted.

☞ Structural analysis of antigen-antibody binding in computational methods can provide a breakthrough in the development of new antibodies. In this paper, we predicted the binding structure of antigen-antibody and predicted the regions and sites involved in these binding. Correlation between antigen-antibody binding capacity and predicted binding structure could not be obtained. We are going to analyze the X-ray crystallography between KNIH90-F1 and RBD in further study. 

2. Line 80 or 469, how can you say “fastest” or “rapidly and conveniently”?

☞ We removed those phrases.

3. Line 143, an information of expression vector (plasmid name, company, etc) is missing.

☞ We added vectors' name.

4. Line 165, please indicate the details of the pseudovirus. Is this express luciferase, GFP, or others? Also, this information should be included in figure 2.

☞ We used luciferase based pseudovirus assay. We corrected the sentence to describe in detail, “Pseudovirus-neutralization assay (for EMC) with luciferase activity [15] and PRNT assay (for EMC and MERS-CoV/KOR/KNIH/002_05_2015 (KNIH-002) strains) were performed as described previously [26]”

5. Line 261, “as this domain is relatively conserved…” seems to be incorrect. S1 positive clones were selected in the B cell sorting as described in materials and methods (line 113). The same thing seems to be applied to line 429 “no S2-specific antibody was isolated…”.

☞ We agreed with you, we corrected the sentence not to make confusion. S2 domain is more conserved than S1 but we did not get any S2 specific antibody. Therefore, it seems to be less immunogenicity.

6. Line 283, table 1 does not indicate the data of “no difference between IC50 values of MERS-CoV EMC and KNIH-002.”

☞ We removed that sentence. 

7. Line 285, “change” should be “differences”.

☞ We corrected it.

---

## [Decision Letter · Decision Letter 1]

22 Apr 2020

Characterization of a human monoclonal antibody generated from a B-cell specific for a prefusion-stabilized spike protein of Middle East respiratory syndrome coronavirus

PONE-D-19-34811R1

Dear Dr. Lee,

We are pleased to inform you that your manuscript has been judged scientifically suitable for publication and will be formally accepted for publication once it complies with all outstanding technical requirements.

With kind regards,

Stefan Pöhlmann, Ph.D.

Academic Editor

PLOS ONE

Additional Editor Comments (optional):

Reviewers' comments:

Reviewer's Responses to Questions

**Comments to the Author**

1. If the authors have adequately addressed your comments raised in a previous round of review and you feel that this manuscript is now acceptable for publication, you may indicate that here to bypass the “Comments to the Author” section, enter your conflict of interest statement in the “Confidential to Editor” section, and submit your "Accept" recommendation.

Reviewer #1: All comments have been addressed

Reviewer #2: All comments have been addressed

Reviewer #3: All comments have been addressed

2. Is the manuscript technically sound, and do the data support the conclusions?

Reviewer #1: Yes

Reviewer #2: (No Response)

Reviewer #3: Yes

3. Has the statistical analysis been performed appropriately and rigorously? 

Reviewer #1: Yes

Reviewer #2: (No Response)

Reviewer #3: Yes

4. Have the authors made all data underlying the findings in their manuscript fully available?

Reviewer #1: Yes

Reviewer #2: (No Response)

Reviewer #3: Yes

5. Is the manuscript presented in an intelligible fashion and written in standard English?

Reviewer #1: No

Reviewer #2: (No Response)

Reviewer #3: Yes

6. Review Comments to the Author

Reviewer #1: (No Response)

Reviewer #2: (No Response)

Reviewer #3: Re: PONE-D-19-34811R1

The concerns in the first manuscripts were well addressed by the revision.

Minor points:

1) line 229-241, methodology to generate standard RNA should be moved to the section “Real-time RT-PCR for virus detection”.

2) line 413, the data is not “time-dependent manner”, since monocyte infiltration was observed in lung at 4dpi. Also, the word “time-dependent manner” is usually used for quantitative data in the time-course experiments. The sentence can be omitted.

7. PLOS authors have the option to publish the peer review history of their article (what does this mean?). If published, this will include your full peer review and any attached files.

Reviewer #1: No

Reviewer #2: No

Reviewer #3: No

---

## [Editor Report · Acceptance letter]

28 Apr 2020

PONE-D-19-34811R1 

Characterization of a human monoclonal antibody generated from a B-cell specific for a prefusion-stabilized spike protein of Middle East respiratory syndrome coronavirus 

Dear Dr. Lee:

I am pleased to inform you that your manuscript has been deemed suitable for publication in PLOS ONE. Congratulations! Your manuscript is now with our production department. 

With kind regards,

on behalf of

Prof. Stefan Pöhlmann 

Academic Editor

PLOS ONE